# Diversity of Legumes in the Cashew Agroforestry System in East Timor (Southeast Asia)

**DOI:** 10.3390/foods11213503

**Published:** 2022-11-03

**Authors:** Lara Guterres, Maria Cristina Duarte, Silvia Catarino, Guilherme Roxo, João Barnabé, Mónica Sebastiana, Filipa Monteiro, Maria Manuel Romeiras

**Affiliations:** 1LEAF—Linking Landscape, Environment, Agriculture and Food Research Center, Associated Laboratory TERRA, Instituto Superior de Agronomia, Universidade de Lisboa, Tapada da Ajuda, 1349-017 Lisbon, Portugal; 2Faculdade de Ciências Exatas, Universidade Nacional Timor Lorosa’e, Avenida Cidade de Lisboa, Dili 314, Timor-Leste; 3cE3c—Center for Ecology, Evolution and Environmental Change & CHANGE—Global Change and Sustainability Institute, Faculdade de Ciências, Universidade de Lisboa, Campo Grande, 1749-016 Lisbon, Portugal; 4Forest Research Center (CEF), Associated Laboratory TERRA, Instituto Superior de Agronomia, Universidade de Lisboa, Tapada da Ajuda, 1349-017 Lisbon, Portugal; 5BioISI—Biosystems and Integrative Sciences Institute, Faculdade de Ciências, Universidade de Lisboa, Campo Grande, 1749-016 Lisbon, Portugal

**Keywords:** food plants, Fabaceae, Southeast Asia, mineral composition, food security

## Abstract

Cashew is an important export-oriented crop in several tropical countries, often under monocropping systems. Intercropping with legume species is promoted as a sustainable practice, enhancing agricultural productivity and providing nutritional food sources to rural communities. This study aimed to characterize the diversity of Leguminosae (or Fabaceae) in the cashew agroforestry systems of East Timor (Southeast Asia). Fourteen cashew orchards were sampled across the country, and information about leguminous species uses was collected from local populations. About 50 species are commonly part of the country’s cashew agroforestry system, many of them simultaneously used as food, fodder, and in traditional medicine. Six bean species—*Cajanus cajan* (L.) Huth, *Phaseolus lunatus* L., *Phaseolus vulgaris* L., *Vigna angularis* (Willd.) Ohwi and H.Ohashi, *Vigna radiata* (L.) R.Wilczek and *Vigna unguiculata* (L.) Walp.—are largely used as food. The mineral contents of these beans revealed relevant differences between species and, in some cases, between types (seed colour) within species. Periods of hunger and low food variety are frequent in East Timor, reflecting a very poor nutritional state of the population. Knowing and using legumes for local nutrition, as well as for healthcare and well-being, adds great value to these species as components of East Timor cashew agroforestry systems.

## 1. Introduction

Southeast Asia (i.e., Brunei, Cambodia, Indonesia, Laos, Malaysia, Myanmar, Philippines, Singapore, Thailand, East Timor, and Vietnam) relies on vulnerable and complex international food supply chains. The consequences of a food security crisis are very real, and as the COVID-19 pandemic weakens and societies return to some kind of normality, the region must adopt a new food security paradigm that can endure internal and external disruptions [1]. With such vulnerabilities, Southeast Asian countries need to better understand the aspects related to food security and climate change while creating conditions that can better harness the productivity of farmers, protect vital migrant labour, increase crop yields, keep borders open, and ensure the sustainability of food production [2].

Situated in this region, East Timor is one of the youngest nations in the world. Since its independence in 2002, East Timorese main activities have been agriculture, fishing, and forestry, and the country’s agriculture share of GDP decreased from 25.6% in 2010 to 15.4% in 2020 [3]. Agriculture is the primary activity, providing subsistence to about 80% of the population. About 225,000 ha of the territory are cultivated, of which 165,000 ha are arable land and 60,000 ha are permanent fields for rice, maize, cassava, coffee, coconut, and other industrial crops [4]. The current agricultural landscapes of the country primarily result from the food production mechanisms implemented by the Portuguese in the 20th century, who promoted rice in coastal areas and coffee at higher altitudes until the 1970s [5]. Most farmers practice subsistence farming where sweet potatoes, cassava, pumpkins, beans, bananas, and vegetables are grown, and small animals such as chickens, goats, and pigs are raised [6].

The country is not self-sufficient in food production and is, therefore, a net food importer [7]. Recent data show that at least 62% of rural households annually experience one to four months of food shortage, and the country’s malnutrition rate, particularly in children under five years of age, is one of the highest in the region [8]. 

The East Timor Strategic Development Plan 2011–2030, implemented by the Ministry of Agriculture, Forestry and Fisheries [9], sets out the priorities to promote economic growth in rural areas, to reduce poverty and provide better food security. To increase the economic return of agricultural crops, the government has promoted the development and increase of export trade through intensive production of cash crops from 2011 onwards: valorisation of high-quality organic coffee and exploitation of cashew as a new alternative cash crop.

Cashew agroforestry systems combine cashew orchards with legumes, spontaneous vegetation and cattle, with the purpose of developing a more sustainable land use that can improve agricultural productivity and the well-being of the rural communities [10]. Growing legumes in co-association with cashew trees (already traditionally used in East Timor) can improve soil fertility due to their ability to fix atmospheric nitrogen in root nodules [11], increase the organic matter content of the soil (as green manure), and reduce soil erosion and increase water infiltration (as a cover crop) [12,13]. Legumes also improve the feeding of ruminants during the dry season, providing high-quality fodder and forage, both in natural and cultivated pastures [14,15]. The use of legumes in crop rotations reduces the need for chemical fertilizers and helps to mitigate pests and diseases in subsequent crops, thus reducing farming costs and protecting the environment [16]. In addition, they provide firewood and construction materials, and can be used as living fences and shade plants.

The Leguminosae family is widely distributed worldwide, with the highest diversity in tropical and subtropical regions [17]. Legumes are among the primary sources of human food, being essential components of a healthy diet [18,19]. Nutritionally, they are 2–3 times richer in protein than cereals [14], and they provide significant amounts of minerals [20]. *Vigna unguiculata* and *Phaseolus vulgaris*, the main pulse crops on the African continent, are essential sources of minerals and vitamins [21,22]. According to Catarino et al. [23], *Vigna unguiculata* has a higher content of boron, magnesium, sulphur, and zinc, while *P. vulgaris* has more iron, calcium, and copper. Zinc and iron deficiencies have the worst impact on public health, but other minerals, such as calcium and magnesium, are also vital to human health [24].

Along with maize and rice, several species of grain legumes, such as kidney beans, soybeans, and mung beans, form the basis of people’s diet in East Timor [25] and are included in traditional dishes such as the *Batar da’an*, which is made of maize, pumpkin, and mung beans. In addition, legumes are important nutritional sources to offset food insecurity, particularly when there is a shortage of maize or rice, usually between November and February.

Adding to their contribution to food security, they play a significant role in the income of resource-poor farmers in many developing countries. In East Timor, farmers have higher incomes when they produce legumes and are thus not dependent on cashew nuts only, which are harvested once a year. The importance of traditional cashew agroforestry systems in East Timor is well-known [26], but knowledge of legume diversity and its significance for the well-being of local populations is missing.

The present work has three main goals: (i) to characterise the diversity of legume species intercropping in the cashew agroforestry systems of East Timor and to evaluate their multiple uses; (ii) to analyse the diversity and mineral composition (macro and microminerals) of some grain legumes (beans); and (iii) to discuss the sustainable use of legumes in the context of food security in East Timor.

## 2. Materials and Methods

### 2.1. Studied Area 

East Timor (Figure 1), with 15,007 km^2^, is a country located on a small island of the Malay Archipelago, which includes six other countries (Indonesia, Malaysia, Philippines, Papua New Guinea, Singapore, and Brunei). This region is recognised for its high plant biodiversity, with about 42,000 vascular plant species, 70% of which are endemic [27,28,29]; the “Malay Archipelago” belongs to the Indo–Malayan transition zone, corresponding to the boundary between the Oriental and Australian biogeographic regions [30]. Influenced by its location in the Wallacean biodiversity hotspot, the flora of Timor Island consists of about 32,500 vascular plant species [29].

Occupying the north-eastern part of Timor Island, East Timor presents a tropical climate, with one wet season (November to April) in the north and two wet seasons (November to April and May to July) in the central highlands and on the south coast [31]. Mean annual temperatures vary from 25 to 27.5 °C near the coast to 15–17.5 °C in the central mountain range, and the annual rainfall varies from <800 mm on the north coast to 2400–2600 mm in the mountains [31]. With a marked topography and reaching an altitude of ca. 2960 meters, the country is characterised by a great diversity of natural ecosystems, such as woodlands, forests, and coastal heaths [32].

According to the East Timor Agriculture Census 2019 [33] and to FAOSTAT 2022 data [34], agricultural holdings in the country cover a total area of 216,189 ha, the top three productions being coffee 36,626 ha, maize 29,455 ha, and rice 20,719 ha. Cashew occupies about 800 ha and bean crops about 10,505 ha, with *Phaseolus vulgaris* being one of the most cultivated (9152 ha) and *Phaseolus lunatus* one of the least cultivated (264 ha).

### 2.2. Diversity of Legumes in Cashew Agroforestry Systems

#### 2.2.1. Sampling

Fourteen cashew agroforestry systems were selected in six East Timor districts: Baucau (2—Baucau, Triloca), Bobonaro (4—Batugadé, Maliana-Maumali, Sanirin, Tunu-bibi), Cova Lima (3—Ai-oan, Beseuc, Suai), Lautém (1—Lospalos), Manatuto (3—Cribas, Manatuto, Natarbora) and Manufahi (1—Fatucahi) (see Figure 1). The legume species present in each one of the studied agroforestry systems were sampled during field surveys between 2018 and 2019. The surveys covered, as far as possible, the whole area of each sampled orchard. Local inhabitants, with previous oral informed consent, were asked about plant species uses (e.g., food, fodder/forage, and medicine) and common names through unstructured interviews. Only leguminous species with traditional uses were considered. Bean species proved to be an important component in cashew agroforestry systems, and six species were sampled in different locations: *Cajanus cajan*—pigeon pea (Batugadé and Beseuc); *Phaseolus lunatus* — lima bean (Kefamenanu and Fatucahi); *Phaseolus vulgaris* — kidney bean (Ai-oan Suai); *Vigna angularis*—azuki bean or rice bean (Baucau); *Vigna radiata*—mung bean (Fatucahi); and *Vigna unguiculata*—cowpea (Fatucahi) (Table 1). 

In each location corresponding to a single cashew orchard, beans were randomly sampled, avoiding the collection of samples from pods of the same plant; this procedure was often difficult because plants can present very long and usually tangled stems. In some cases, more than one species of beans is cultivated in the same orchard (e.g.; Fatucahi—*Phaseolus lunatus*; *Vigna radiata* and *Vigna unguiculata*).

A total of 144 grain samples were analysed, including 6 species of beans and 12 colour types (more details in Table 1 and Figure 2). To the species presenting mixed colour-type beans in the same sampled location (e.g., *Cajanus cajan* and *Phaseolus lunatus*), the different colour types were later separated in the laboratory; the very few collected from some colour types (e.g., the “cream and purple” type of *Phaseolus lunatus*) accounts for the different number of samples (Table 1). In addition, these two species were collected in two different locations in order to have a 10 g minimum sample weight (see point 2.3) of the least represented colour types.

During the fieldwork, information on the cultivation of various types of beans in East Timor was also obtained from local people.

#### 2.2.2. Database Construction 

The data obtained during the field surveys were used to build a database with all the Leguminosae species associated with cashew agroforestry systems in East Timor. This database includes, for each species, the scientific and common names, growing habit (tree, shrub or climber), native distribution, naturalness in East Timor, and main uses: food, fodder/forage, medicine, and domestic use. Information on taxonomic data and native distribution was obtained from online databases (e.g., Plants of the World Online [35] and International Legume Database and Information Service [36]). 

### 2.3. Mineral Analyses

Mineral analyses were carried out separately for seeds of each species or each colour type (in the case of species with more than one colour type) and for each site (for species collected from more than one site). Three replicates were made per sample, each with about 10 g of seeds separated and ground into flour using a coffee grinder (Kunft). 

Concentrations of 11 minerals (boron—B, calcium—Ca, copper—Cu, iron—Fe, potassium—K, magnesium—Mg, manganese—Mn, sodium—Na, phosphorous—P, sulphur—S, zinc—Zn) in seeds of *Cajanus cajan*, *Phaseolus lunatus*, *Phaseolus vulgaris*, *Vigna angularis*, *Vigna radiata*, and *Vigna unguiculata* were determined by inductively coupled plasma optical emission spectrometry (Thermo Scientific iCAP 7000 Plus Series ICP-OES spectrometer, Thermo Fisher Scientific, Waltham, MA, USA) after digestion in aqua regia (a mixture of nitric acid and hydrochloric acid (1:3)) at 105 °C during 60 min [23,37]. Calibration was made using the controls prepared with the same analytical procedure and reagents but without sample. Calibration curves of at least five different concentrations were used to quantify each element. The analyses were performed in triplicate and the results were expressed as mg per L. The conversion to mg per kg wet weight (ww), represented by Z, was done using the following formula:

$$Z\;\left(\mathrm{mg}/\mathrm{kg}\right)=L\times \frac{50}{P}$$where *L* is the reading value given in mg/L and *P* is the sample weight in g; 50 mL is the total volume of the analysed solution.

### 2.4. Statistical Analysis

Descriptive statistics were calculated for each species and each colour type, namely mean and standard deviation of mineral content (mg/Kg). Boxplots were also constructed to understand the variation in mineral content between samples.

Since some mineral contents did not follow a normal distribution (*p* < 0.05 with the Shapiro–Wilk test [38], we performed non-parametric tests. For comparison between two or three groups, the Mann–Whitney and the Kruskal–Wallis (respectively) tests were chosen. For the latter test, when the null hypothesis was rejected (*p*-value < 0.005), the Dunn-test of the FSA R package was applied [39], which includes the Bonferroni-type adjustment of *p*-values to reduce the probability of committing a type II error, ensuring a relatively high level of statistical power.

The samples (*n* = 144) were classified based on a cluster analysis using the Euclidean distance as a measure of similarity between the mineral contents of the various species. The clustering method used was the “average”, using the “*hclust*” function of the “*vegan*” package [40].

Data standardisation (mean = 0 and standard deviation = 1) was the first step required to perform a principal component analysis (PCA). The multivariate analysis by PCA, based on the correlation matrix, and the eigenvectors and eigenvalues, were projected and visualised with the “*ggplot*” function of the “*ggplot2*” package [41]. All analyses were conducted in the R program [42].

## 3. Results

### 3.1. Diversity of the Legume Species in Cashew Agroforestry Systems

Fifty Leguminosae species were identified in the studied cashew agroforestry systems (Figure 3; Table 2), and most of them are commonly cultivated exotic species.

Trees represent half of the species (25), with *Acacia* and *Erythrina* being the most common genera; 15 species are shrubs, including crops such as pigeon pea (*Cajanus cajan*), soybean (*Glycine max*) and peanut (*Arachis hypogaea*). The herbaceous species (10) represent only a minor proportion of the recorded legumes; all of them are vines and include several species of grain legumes (such as beans).

Most of the species are used by local populations as medicine (36 species, 72%) or as forage or fodder/forage (34 species, 68%). Twenty-three species (46%) were reported as food and 13 species (26%) as materials for domestic uses (mainly wood). Five species were simultaneously reported by the locals to all four use classes: *Gliricidia sepium*, *Leucaena leucocephala*, *Senna siamea*, *Sesbania grandiflora*, and *Tamarindus indica*. *Gliricidia sepium* produces edible flowers, and the leaves are an important food source for cattle. The species is frequently harvested for firewood and is also used in traditional medicine [35,43]. *Leucaena leucocephala* is mainly used for cattle food, but the seeds could also be used for cooking [44]. *Senna siamea* fruits and leaves can be eaten as a vegetable; it is also used as a shade tree, forage, and medicinal plant [43]. *Sesbania grandiflora* is used as firewood, and for house building and food in rural communities. It is an important species to feed the ruminants, as medicine, organic fertilizer, and ornamental plant [35,44]. *Tamarindus indica* produces tasty fruits very much employed in human consumption and its timber is proper for house building [35]. This species is suitable to feed animals and for medicinal purposes against diarrhea, constipation, fever, and other diseases [44].

Moreover, a large group, including *Arachis hypogaea*, *Caesalpinia pulcherrima*, *Cajanus cajan*, *Calopogonium mucunoides*, *Glycine max*, *Mucuna pruriens*, *Phaseolus lunatus*, *Senna occidentalis*, *Senna tora*, *Vigna radiata*, and *Vigna unguiculata*, has food, fodder/forage and medicinal uses. Most species classified for domestic uses were applied for construction, furniture manufacture or firewood, but two were reported for other uses: dye (*Indigofera suffruticosa*) and detergent (*Samanea saman*) [35,43,44].

Among the most frequent leguminous species, the multipurpose *Gliricidia sepium*, present in seven sites from four districts, was clearly the most reported species, followed by *Indigofera suffruticosa* in five sites; *Leucaena leucocephala*, *Moringa oleifera*, *Vachellia farnesiana*, *Vigna unguiculata*, and *Vigna angularis* in four sites; and *Acacia ancistrocarpa*, *Acacia iteaphylla*, *Acacia stenophylla*, *Delonix regia*, *Samanea saman*, and *Sesbania grandiflora* in three sites.

Cribas is the sampled area with the highest diversity of legume species, with 23 different species corresponding to 46% of the total number. Batugadé has 16 species (32%), Natarbora has 14 species (28%), and Maumali, Sanirin, Fatucahi have 6 species each (12%). The other sampled areas have less than 10% of the recorded legume species. Overall, the Manatuto and Bobonaro districts present the highest diversity values, with 31 and 28 species, respectively; together, these two districts account for 86% of all the reported leguminous species.

### 3.2. Legume Beans in Cashew Agroforestry Systems

Among the alimentary species usually present in East Timorese cashew agroforestry, several grain legumes, particularly bean species, are cultivated: pigeon pea (*Cajanus cajan*), lima bean (*Phaseolus lunatus*), kidney bean (*Phaseolus vulgaris*), azuki bean (*Vigna angularis*), black gram (*Vigna mungo*), mung bean (*Vigna radiata*), and cowpea (*Vigna unguiculata*) (Table 2); most of them also have other uses (namely, as fodder and medicinal). 

As with other Leguminosae species present in these agrosystems, most grain legumes are of exotic origin: *Phaseolus lunatus* and *Phaseolus vulgaris*—from Central America; *Vigna unguiculata*—from West and Central Africa; *Cajanus cajan* (Figure 4) and *Vigna angularis*—from Asia; and *Vigna mungo*—from the Indian subcontinent. *Vigna radiata* is the only native species in East Timor, with a distribution extending from the Arabian Peninsula to Asia and Australia.

In addition to the surveyed locations, bean species are reported as commonly cultivated in other East Timorese cashew agroforestry systems (see Table 2), from low altitudes to about 1600 m, as is the case of *Phaseolus vulgaris*, *Cajanus cajan*, *Vigna unguiculata*, *Vigna angularis* and *Phaseolus lunatus*; in contrast, *Vigna radiata* is found only up to 50 m. This species and *Vigna angularis* have more restricted occurrences, while *Phaseolus vulgaris*, *Cajanus cajan*, and *Vigna unguiculata* are the most widespread species.

According to information locally obtained, kidney bean (*Phaseolus vulgaris*) is the most popular bean in East Timor. It presents several seed colour types in Timor (red, black, black and red, brown, white, mixed cream and brown), with each place having its own type, different in flavour and physical characteristics (e.g., hard or soft when cooked). In East Timor, the seeds commonly sold in the market are of mixed types and are also likely to be mixed for sowing, in part to reduce the incidence of pests and diseases. Particularly well-adapted to high, cool areas, beans require adequate soil moisture during establishment, vegetative growth and flowering, but prefer drier climatic conditions during fruit development and ripening. The great distribution and diversity of kidney bean demonstrate that it is extremely well-adapted to the climatic conditions of East Timor since this is one of the most climatically demanding bean species [45]. This is the only bean species cultivated in high and cold areas, where rainfall is more regular and air humidity is higher. Cultivation at altitudes from 800 to 1600 m (not irrigated) produces better quality beans and higher yields. Usually cultivated at the same time as maize, it is extensively planted in several areas (e.g., Baguia, Quelicai, and Fatumaca).

The lima bean (*Phaseolus lunatus*) presents three different colour types (purple and white, cream and purple, and brownish). It is used for medicinal purposes, food, and forage; however, the use for human consumption is decreasing due to the cooking requirements.

Pigeon pea (*Cajanus cajan*) is also very frequent in East Timor, with several types (cream, black, brown, and mixed cream and brown seeds). In addition to rice, pigeon pea is one of the main pre-European Timorese crops. Together with mung bean, *Cajan cajan* is particularly important in the subsistence regimes of local populations and can be widely found in fields and gardens. Due to its resistance to drought, pigeon pea is highly important in local agriculture, being often available when other crops that critically depend on rainfall fail, such as rice and maize, as reported by Fox [5]. *Cajanus cajan* can be cultivated in sandy soils, but the productivity is higher in soils with more organic matter. In Ataúro, the most productive place, larger pods are observed.

Cowpea (*Vigna unguiculata*) and mung bean (*Vigna radiata*) are grown in East Timor mainly in home gardens or small areas. *Vigna unguiculata* presents several types (black, brownish red, yellow, cream or white seeds). It is usually sown together with maize and cassava, which serve as support for the growing beans. In East Timor, cowpea has proven to be an important crop for rural populations, being more frequently grown at low altitudes, while mung bean is grown in regions of more fertile soil and has a lower diversity of types. Mung bean (*Vigna radiata*) is well adapted to more clayey and muddy soil, where productivity is higher. It is traditionally sown in agroforestry systems (intercropping) in Cova Lima, Fatucahi and Betano and also in floodplains after rice harvesting. It is the most expensive bean on the market, followed by *Phaseolus vulgaris*. Mung beans are widely used to feed children and are considered very nutritious.

The azuki bean (*Vigna angularis*), of which several types are cultivated (bordeaux, black, and cream seeds), is well adapted to hot and humid places in East Timor. The highest yields occur in the districts of Baucau and Aileu. It has a typical salty taste and is preferred in daily consumption by local communities and in ritual ceremonies instead of *V. unguiculata*, mainly in typical cuisine (to color rice dishes and in *feijoada*, a traditional dish). In addition to dry seeds, the population consumes green pods as vegetables. This bean has a lower economic value on the market since most people prefer *V. unguiculata*, *V. radiata* or *Phaseolus vulgaris*.

### 3.3. Mineral Composition

The mineral contents were screened in the six selected legume species (Section 3.3.1) and respective seed colour types (Section 3.3.2): *Cajanus cajan* (4 types), *Phaseolus lunatus* (3 types), *Phaseolus vulgaris* (1 type), *Vigna angularis* (1 type), *Vigna radiata* (1 type), and *Vigna unguiculata* (2 types). 

The contents of Na were not quantifiable by the ICP-OES technique due to their low concentrations (lower than 8.333 mg/kg ww) and are therefore not presented in the figures. The same occurred for boron in *Cajanus cajan* (values lower than 1.667 mg/kg ww).

#### 3.3.1. Mineral Profiles of the Six Legume Species 

The mineral contents in the six legume species are presented in Figure 5 and Appendix A (grey rows). Concerning macrominerals, Ca contents ranged from 2916.046 ± 62.443 mg/kg ww in *Vigna angularis*, well above the values presented by the other species, to 420.501 ± 70.300 mg/kg ww in *Phaseolus lunatus*; K contents ranged between 12,770.371 ± 701.176 mg/kg ww in *Phaseolus lunatus* to 5604.853 ± 198.382 mg/kg ww in *Vigna radiata*; as to Mg contents, *Vigna angularis*, with 1777.535 ± 61.257 mg/kg ww, and *Cajanus cajan*, with 763.173 ± 48.714 mg/kg ww, respectively display the highest and lowest values; the highest P content was 4475.780 ± 289.892 mg/kg ww in *Phaseolus vulgaris* and the lowest was 2130.565 ± 229.839 mg/kg ww in *Cajanus cajan*; finally, S content ranged from 1862.081 ± 194.575 mg/kg ww in *Vigna unguiculata* to 1108.567 ± 150.129 mg/kg ww in *Phaseolus lunatus*.

Concerning microminerals (Figure 5 and Appendix A), B ranged from 25.224 ± 5.310 mg/kg ww (*Phaseolus lunatus)* to 2.528 ± 2.232 mg/kg ww (*Vigna radiata*) (note that the values of *Cajanus cajan* were even lower, as it was not possible to quantify them); Cu values ranged from 8.141 ± 0.692 mg/kg ww in *Vigna unguiculata* to 1.947 ± 0.486 mg/kg ww in *Cajanus cajan*; Fe values ranged from 63.326 ± 9.033 mg/kg ww in *Phaseolus vulgaris* to 17.158 ± 5.142 mg/kg ww in *Cajanus cajan*; *Vigna angularis* presented the highest Mn contents 21.091 ± 5.435 mg/kg ww and *Cajanus cajan* 1.684 ± 0.800 mg/kg ww the lowest; finally, Zn values varied between 29.564 ± 4.395 mg/kg ww in *Vigna unguiculata* and 12.760 ± 1.294 mg/kg ww in *Cajanus cajan*. For all micronutrients, except Cu and Zn, the species exhibiting the highest values show significant differences.

The cluster analysis (see Figure 6) clearly separated the different genera (i.e., *Cajanus*, *Phaseolus*, *Vigna*) in terms of mineral contents; *Cajanus cajan* is more similar to *Phaseolus* spp. than to *Vigna* spp. The Principal Component Analysis (PCA) (see Figure 7; Appendix A) also clearly showed the specific profiles in mineral contents of the analysed species. *Vigna angularis* and *V. unguiculata* presented the highest contents in minerals, while *Cajanus cajan* presented the lowest values. The first four principal components accounted for 95.33% of the variance among the six species; PC1 and PC2 accounted for 77.3% of that variability, with Zn, Mg, and Cu being the most important contributors, with positive coefficients (PC1), and B with a negative coefficient (PC2). PC3 showed Ca as a strong positive contributor and P as a negative one.

#### 3.3.2. Mineral Profiles Variability in Seed Colour Types

The mineral contents were independently screened for the species with more than one seed colour type—*Cajanus cajan* (4 types), *Phaseolus lunatus* (3 types), and *Vigna unguiculata* (2 types) (Figure 8, Figure 9 and Figure 10, Appendix A, white rows).

The “cream” *Cajanus cajan* (Figure 8) exhibited a S content of 1219.653 ± 40.269 mg/kg ww and a Zn content of 13.865 ± 0.915 mg/kg ww, values that are significantly different from the other colour types. The “cream-brownish/purple” type also presented significant differences concerning Mn 1.992 ± 1.156 mg/kg ww and P contents 2537.865 ± 90.150 mg/kg ww. 

In *Phaseolus lunatus* (Figure 9) the colour type “cream and purple” presented the highest values of S, Cu, Fe, K, Mg, P, Zn; the mineral contents only differed significantly from “purple-white” colour type, this type presenting the lowest values for most nutrients. The K value of 13519.906 ± 223.489 mg/kg ww presented by the “cream and purple” colour type is the highest value of all the analysed samples (irrespective of the species).

Finally, the “bordeaux” *Vigna unguiculata* (Figure 10) had values of Ca, Mg, and Mn significantly higher than those presented by the “black” colour type. This species displayed the overall highest values of Cu (8.251 ± 0.579 mg/kg ww, “black” colour type), S (1990.741 ± 88.192 mg/kg ww, “black” colour type), and Zn (29.601 ± 5.731 mg/kg ww, “bordeaux” colour type). 

When considering the different seed colour types per species, PCA (Figure 11) and cluster analysis (Figure 6) show different situations. In the case of *Cajanus cajan*, colour types (“cream”, “black”, “brownish and cream” and “brown”) can be fairly distinguished in the PCA (Figure 11a; Appendix A); the four principal components accounted for 81.91% of the variability, with PC1 and PC2 accounting for 55.75% of that variability. With positive coefficients, zinc and sulphur were the main contributors to PC1, and phosphorus was the most important contributor to PC2, also with a positive coefficient. A large mineral diversity distinguished the four colour types, except for the very similar black and brownish. The geographical origin of the accessions (Batugade and Beseuc) also seems to account for some differences, with Beseuc samples (presenting the highest contents in minerals) mainly in the right part of the plot. Similar results were obtained in cluster analysis (Figure 6), with samples from the same provenance tending to cluster together. 

In the case of *Phaseolus lunatus* (“purple and white”, “cream and purple”, and “brownish”) (Figure 11b; Appendix A), the four principal components accounted for 92.57% of the variability, with PC1 and PC2 accounting for 70.31%. Phosphorus and calcium, both with negative coefficients, were the most important contributors to PC1 and PC2, respectively. The different colour types presented clearly different mineral profiles. Note that each group included samples from two geographical origins (Kefamenanu and Fatucahi). In cluster analysis (Figure 6), “purple and white” mineral profiles were similar irrespective of the origin, whereas no significant differences between “brownish” and “cream and purple” or their origins were found. 

Regarding *Vigna unguiculata* (“black” and “bordeaux” types) (Figure 11c; Appendix A), the four principal components accounted for 83.60% of the variability, with PC1 and PC2 accounting for 57.09%. The most important contributors for PC1 and PC2 were, respectively, zinc and sulphur, both with positive coefficients. The mineral contents allowed the distinction of the two colour types, which are also of different geographical origins. A similar result was obtained in the cluster analysis (Figure 6).

## 4. Discussion

Growing concerns about environmental and food security make it necessary to implement more sustainable agricultural practices, which are a major challenge for East Timor farmers [46]. This study revealed, for the first time, the diversity of legumes associated with the cashew agroforestry systems and contributes with new data to a sustainable strategy for reducing micronutrient deficiencies still found in the East Timor populations [47]. Furthermore, legumes used as intercrops in cashew plantations enrich the soil with fixed atmospheric nitrogen and reduce the need for mineral fertilizers, thus contributing to the sustainability and economic viability of the cropping system [48].

Intercropping in cashew is mainly practiced in young orchards until they reach fructification (i.e., 2–3 years). This system generates income until the revenue from cashew can be obtained, but it is also an effective weed control method during the early stages of cashew field establishment [49]. The presence of several crop legumes such as cowpea, groundnut, soybean, mung bean, and pigeon pea in cashew plantations is also common in other tropical countries, namely in Nigeria [50] and India [51].

In East Timor, we identified 50 different legume species with ecological and economic importance growing in cashew orchards. Most of them (72%) are used as medicinal plants, 68% are used as forage, 46% as human food and 26% as materials for domestic uses. The application of the Leguminosae family in traditional medicine is widely recognized in other Southeast Asian countries [52,53,54]. These species are an invaluable resource for rural populations in developing countries with weak health systems, such as East Timor. Leguminous species used for human consumption include: groundnuts (*Arachis hypogaea*), for cooking, frying, or roasting; soybeans (*Glycine max*), to make “tempeh” and “tofu”; peas (*Lathyrus oleraceus*) and the seeds of *Senna occidentalis* and *Senna tora*, roasted, as a coffee substitute; or even the edible flowers of *Delonix regia* and *Sesbania grandiflora*. The seeds of *Canavalia gladiata* and *Leucaena leucocephala* are also consumed by people living in rural areas of East Timor, although these species require some precautions during their preparation because of their toxicity and need to be boiled in three changes of water to eliminate toxic compounds [55]. Despite its nutritional potential in terms of protein content, these species are not commonly used as a food or cultivated like other legumes, due to the presence of antinutritional factors such as haemagglutinins (concanavalin A), protease inhibitors, hydrocyanic acid, tannins, phytates, and canavanine [56].

Moreover, several legume trees present in the surveyed agroforestry systems are well known for their forage potential (e.g., *Gliricidia sepium*, *Sesbania grandiflora*, *Samanea saman*, and *Leucaena leucocephala*), being widely used in East Timor [57]. To promote cashew as a cash crop in East Timor, the co-association with legume species is important to enhance the diversity and multipurpose of the agroforestry system, particularly when compared with other cashew-producing countries, such as Guinea-Bissau or Mozambique [58,59], where the monocultural regime is practiced, favouring the outbreak and/or expansion of pests and diseases that reduce productivity.

The present study provides the phytochemical characterization of six bean species (i.e., *Cajanus cajan*, *Phaseolus lunatus*, *P. vulgaris*, *Vigna angularis*, *V. radiata*, and *V. unguiculata*) associated with cashew agroforestry systems in East Timor. These species are the basis of local population’s diets (Figure 12) and are also used for fodder or forage. In addition, medicinal uses were reported for some species (*Cajanus cajan*, *Phaseolus lunatus*, *Vigna radiata*, and *V. unguiculata*).

Nowadays, the cultivation of *Phaseolus lunatus* in East Timor is very limited since a time-consuming cooking process is required to avoid poisoning (boiled up to 10 times, discarding water after each boiling) [60]. This is one of the few pulses species that contains toxic amounts of cyanide-producing glucosides [61].

Of the six species analysed for their mineral contents, *Vigna unguiculata* is the richest in Cu, S, and Zn; *V. angularis* exhibits the highest content in Ca, Mg, and Mn; *Phaseolus vulgaris* exhibits the highest content in P and Fe; and *P. lunatus* is the richest in B and K. Increasing legumes consumption at the household level can overcome some dietary deficiencies (e.g., in iron and zinc). The presence of iron and zinc in the diet is vital for humans as both are responsible for essential body functions and their deficit can lead to severe medical conditions. Iron is required to transfer oxygen to the body’s tissues and organs, and anaemia is the most common nutritional deficiency, affecting more than 2 billion people worldwide [62]. Zinc plays an essential role in human metabolism, improving the immune system and preventing disease [63]. According to East Timor’s Demographic and Health Survey [64], in 2016, about 40% of the children, 23% of the women aged 15–49, and 13% of the men aged 15–49 were anaemic. According to our results, *Phaseolus vulgaris* and *Vigna unguiculata* are the most iron-rich species, the latter also having a higher zinc content. Thus, by providing East Timor farmers with new data concerning the importance of producing high-nutritional-value beans as a feasible and sustainable solution for the alleviation of malnutrition, their revenue and economic wealth will also increase.

Previous works reported that seed coats are rich in several minerals, e.g., Ca, Mg, Mn, Cu, Zn, B, Al and Na [65]; in addition to variations between species, the minerals present in the seed coat vary widely within the same species [20,66]. For instance, in our study, Ca, Mg, and Mn contents are significantly higher in “bordeaux” than in “black” *Vigna unguiculata* (Appendix A, Figure 10); the importance of the colour of the seed coat is also noticeable with S content but, for this element, it is the “black” type that presents the highest values. *Phaseolus lunatus*’ “purple-white” colour type displays the lower values for several nutrients (Cu, K, Mn, S, Fe, Mg, P, and Zn), whereas “cream-purple” and “brownish” present most of the highest values. Therefore, we support the view that seed coats are relevant to explain the variations in the mineral levels of the seeds. In addition to minerals, previous works reported that seed-coat colour is also correlated with the presence of other components such as flavonoids and tannins [67,68]. Many of these components have antioxidant properties, and a clear correlation between antioxidant properties and colour of the seed coat has been demonstrated (e.g., in cowpea seed coat [69,70]). Adedayo et al. [71] found that the more colourful seeds of *Vigna subterranea* have greater antioxidant capacity. In *Phaseolus vulgaris,* antioxidant capacity is higher in beans with dark seed coats, in which phenolic compounds and flavonoids were found in higher quantities [72].

Resulting from the long domestication process induced by humans, the colour of the seed-coat pattern is an essential trait in beans, improving the attractiveness of the seeds. Local types display a remarkable diversity of colours, but this richness often remains undervalued [70]. Indeed, more than just an aesthetic preference, the variation of the mineral composition with seed colour is a relevant issue that remains to be properly studied.

## 5. Conclusions

With the current global emphasis on sustainable agriculture, there are compelling reasons to accelerate the development of legume-based agroforestry systems. Legumes, particularly beans, are extremely important for food and nutritional security in countries such as East Timor, where access to food is scarce and subsistence agriculture is the main activity of the rural population. In fact, the adverse edapho-climatic conditions of this country, characterized by seasonal torrential rains and soil infertility, cause poor local agricultural production. This leads to lack of food products, the need for imports and increased prices, aggravating nutritional insecurity through reduced access to food. Grain legumes are among the most common food crops in East Timor. The present study involved a survey of the diversity of legume species grown in cashew agroforestry system in East Timor, focusing on grain legumes. The assessment of legume diversity and its local uses, as well as the evaluation of the mineral content of a very important group of grain legumes, highlighted their importance for food security in this country. The results revealed that the studied legume species are excellent sources of minerals, representing an invaluable potential to meet the nutritional needs of the Timorese populations. 

Improving food security is one of the main objectives of the Government of East Timor, and the National Strategic Development Plan 2011–2030 mentions the urgent need to improve and train the agricultural sector to reduce poverty, increase food security, and promote the country’s economy [73]. Thus, in East Timor, the leguminous plants in co-association with cashew (cash crop) are important as promoters of diversity in the cashew agroecosystem and also of food and nutritional security. 

## Figures and Tables

**Figure 1 foods-11-03503-f001:**
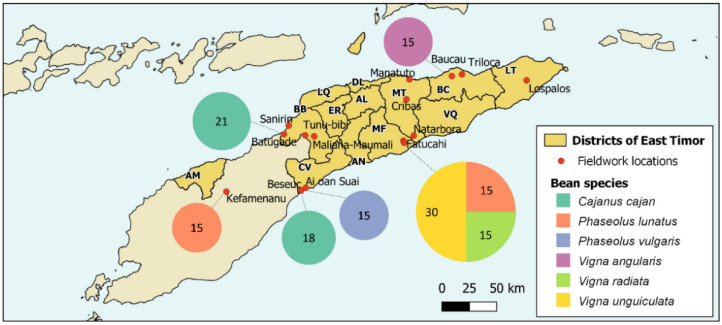
Map of Timor with the districts of East Timor (AL—Aileu; AN—Ainaro; BC—Baucau; BB—Bobonaro; CV—Cova-Lima; DL—Díli; ER—Ermera; LT—Lautém; LQ—Liquiçá; MT—Manatuto; MF—Manufahi; AM—Oecussi-Ambeno; VQ—Viqueque) and the locations of fieldwork. The size of the circles represents the number of samples of beans collected in each location.

**Figure 2 foods-11-03503-f002:**
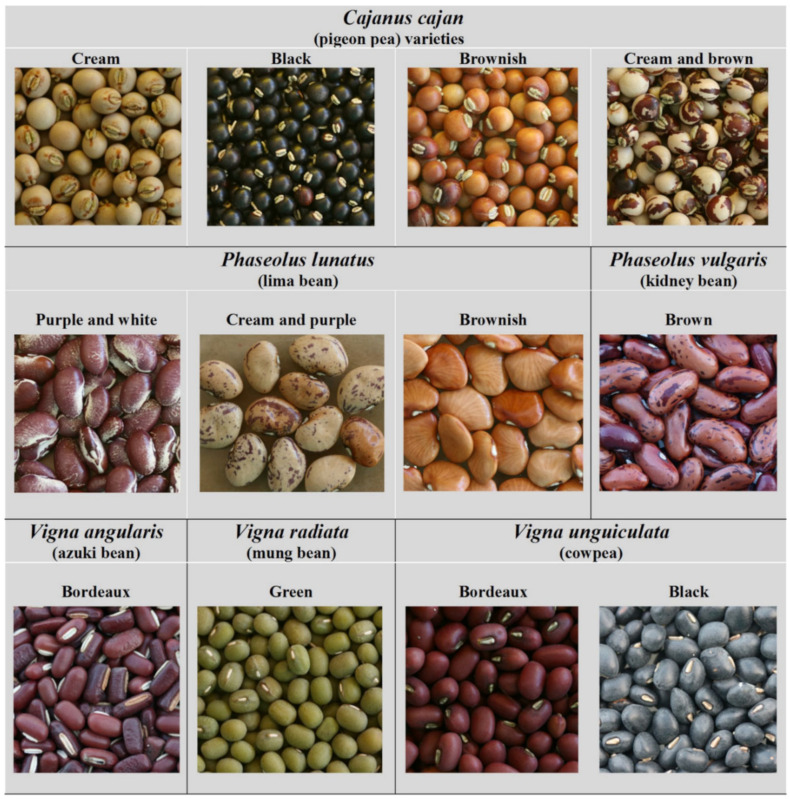
Sampled species and types according to their seed colours.

**Figure 3 foods-11-03503-f003:**
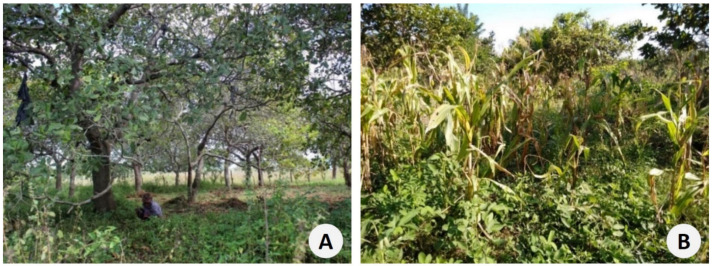
Cashew agroforestry systems in East Timor: (**A**) in Maliana (Maumali): cashew and *Crotaria pallida*; (**B**) in Fatucahi: cashew, maize, beans (*Vigna unguiculata*), and peanut (Photos L. Guterres).

**Figure 4 foods-11-03503-f004:**
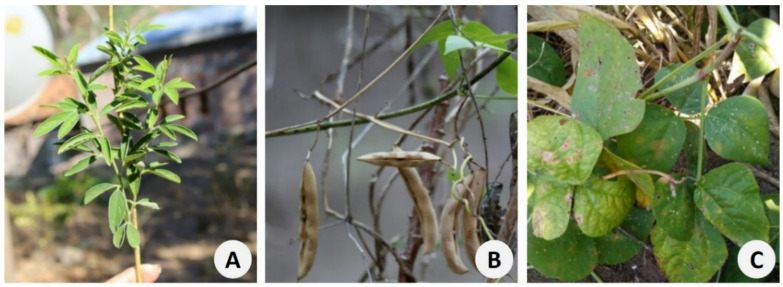
(**A**) *Cajanus cajan*. Kefamenanu (Kupang); (**B**) *Phaseolus lunatus*. Kefamenanu (Kupang); (**C**) *Vigna unguiculata* Beseuc (Cova Lima) (Photos L. Guterres).

**Figure 5 foods-11-03503-f005:**
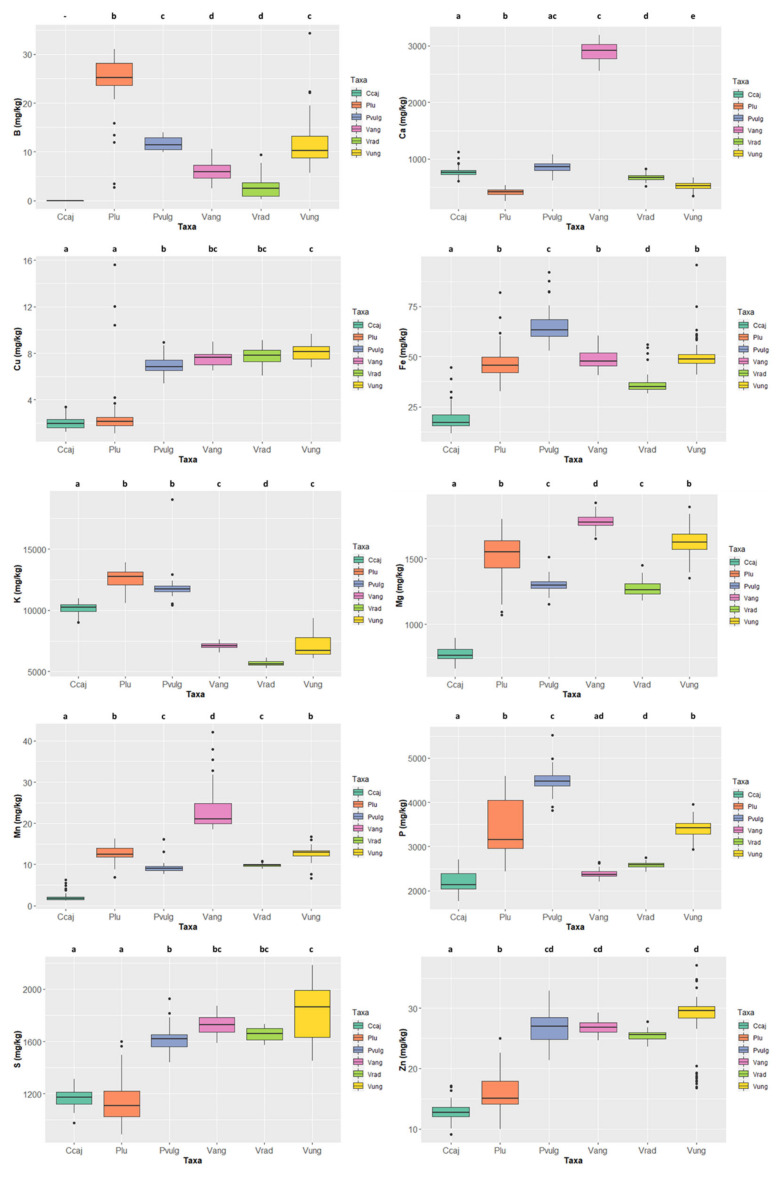
Variation of the content of the 10 minerals quantified in the six species (*Cajanus cajan*, *Phaseolus lunatus*, *Phaseolus vulgaris*, *Vigna angularis*, *Vigna radiata*, and *Vigna unguiculata*). The box represents the 25th, 50th (median), and 75th percentiles, while the whiskers represent the 10th and 90th percentiles with minimum and maximum observations. The dots represent the outliers. Different lower-case letters indicate significant differences (*p* < 0.05) between species.

**Figure 6 foods-11-03503-f006:**
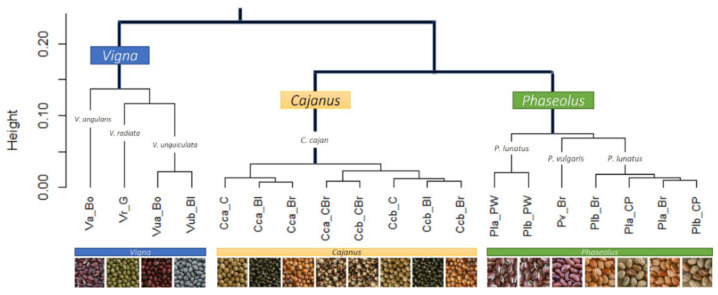
Cluster analysis of the mineral composition of the six legumes and their seed colour types.

**Figure 7 foods-11-03503-f007:**
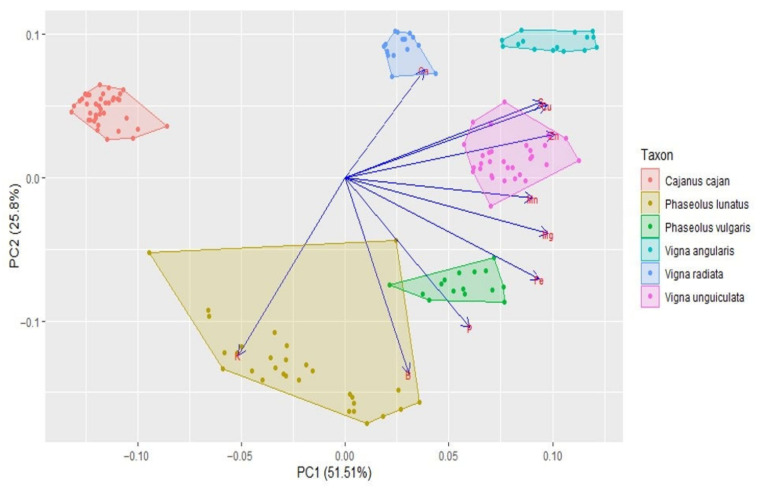
Principal components analysis based on the mineral content of *Cajanus cajan*, *Phaseolus lunatus*, *P. vulgaris*, *Vigna angularis*, *V. radiata*, and *V. unguiculata* (seed colour types pooled together for each species). The length of the arrows represents the differences in the variance explained in relation to the other arrows.

**Figure 8 foods-11-03503-f008:**
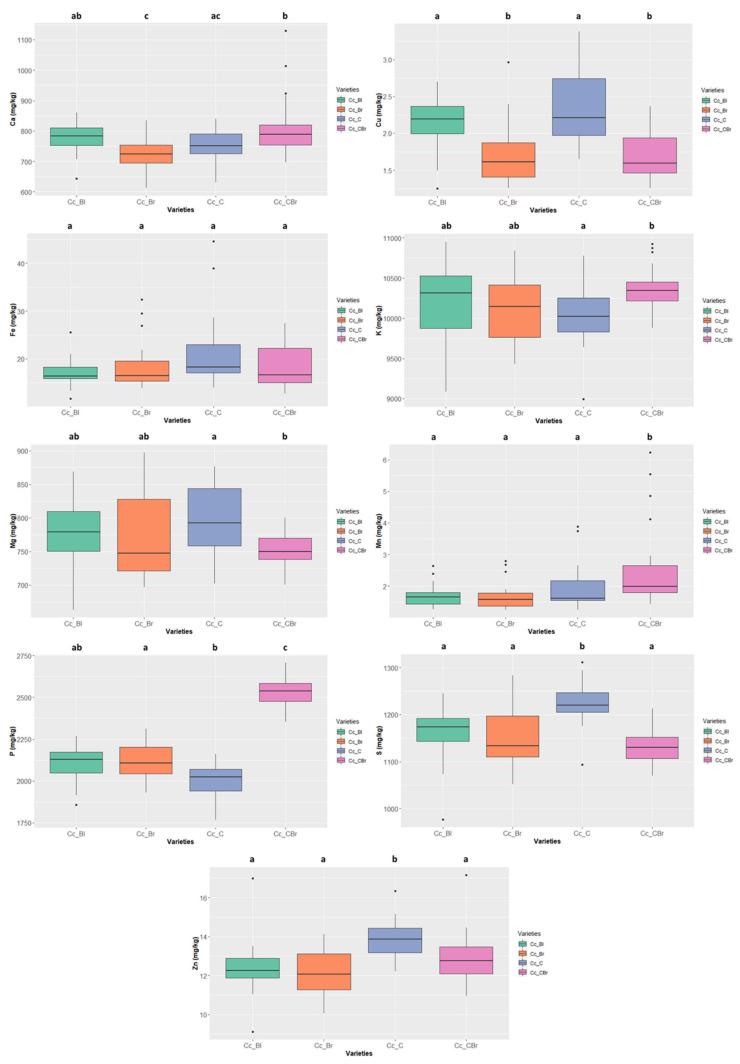
Variation of the content of the 9 minerals quantified in the different colour types of *Cajanus cajan*. The box represents the 25th, 50th (median), and 75th percentiles, while the whiskers represent the 10th and 90th percentiles with minimum and maximum observations. The dots represent the outliers. Different lower-case letters indicate significance differences (*p* < 0.05) between species.

**Figure 9 foods-11-03503-f009:**
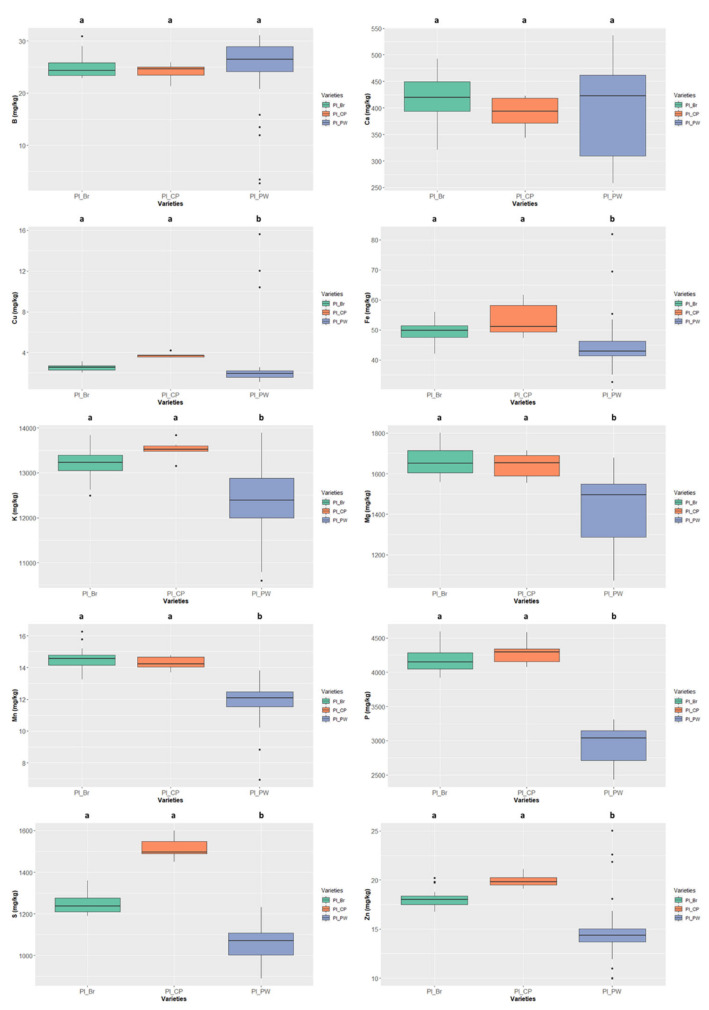
Variation of the content of the 10 minerals quantified in the different colour types of *Phaseolus lunatus*. The box represents the 25th, 50th (median), and 75th percentiles, while the whiskers represent the 10th and 90th percentiles with minimum and maximum observations. The dots represent the outliers. Different lower-case letters indicate significance differences (*p* < 0.05) between species.

**Figure 10 foods-11-03503-f010:**
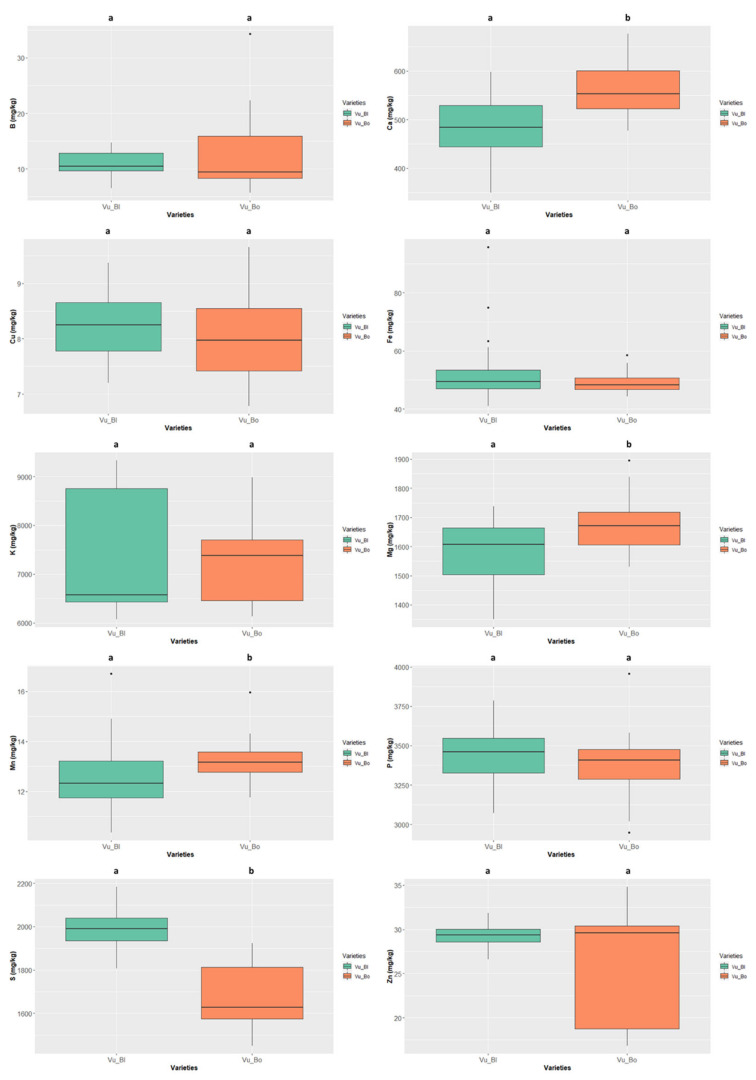
Variation of the content of the 10 minerals quantified in the different colour types of *Vigna unguiculata*. The box represents the 25th, 50th (median), and 75th percentiles, while the whiskers represent the 10th and 90th percentiles with minimum and maximum observations. The dots represent the outliers. Different lower-case letters indicate significance differences (*p* < 0.05) between species.

**Figure 11 foods-11-03503-f011:**
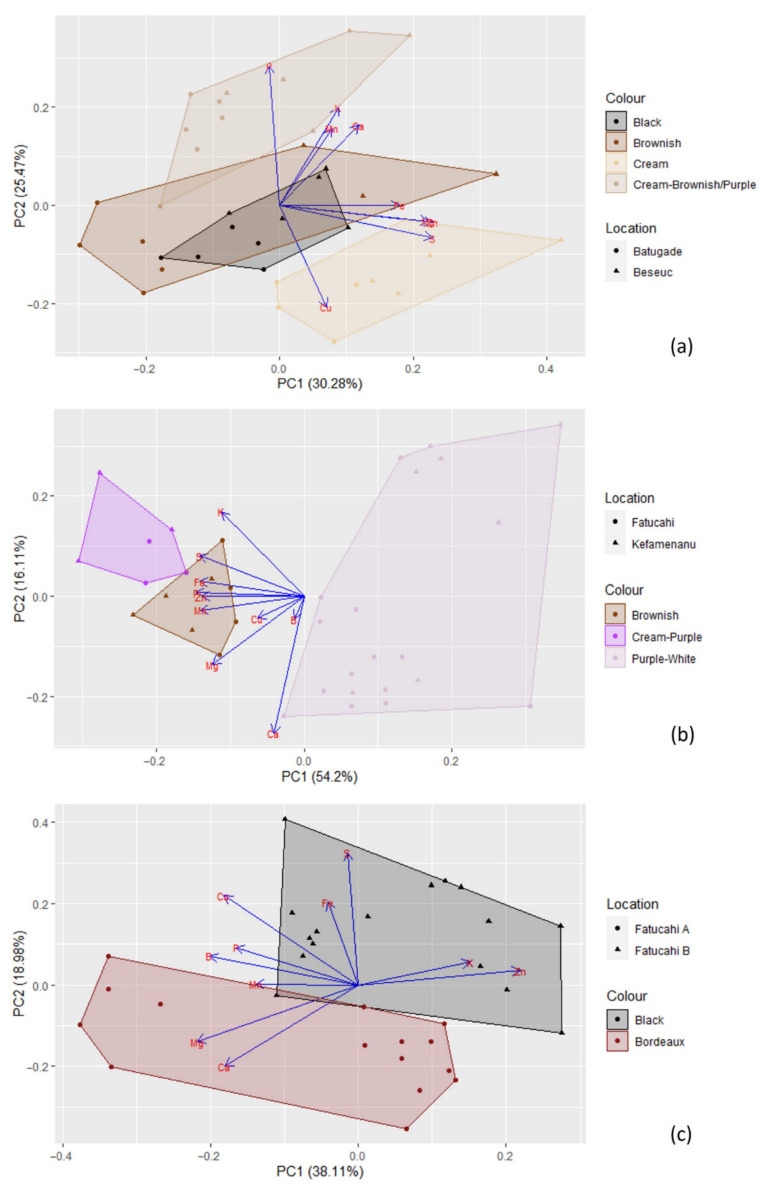
Principal components analysis based on the mineral content of (**a**) *Cajanus cajan* (4 types); (**b**) *Phaseolus lunatus* (3 types); and (**c**) *Vigna unguiculata* (2 types). The length of the arrows represents the differences in the variance explained in relation to the other arrows.

**Figure 12 foods-11-03503-f012:**
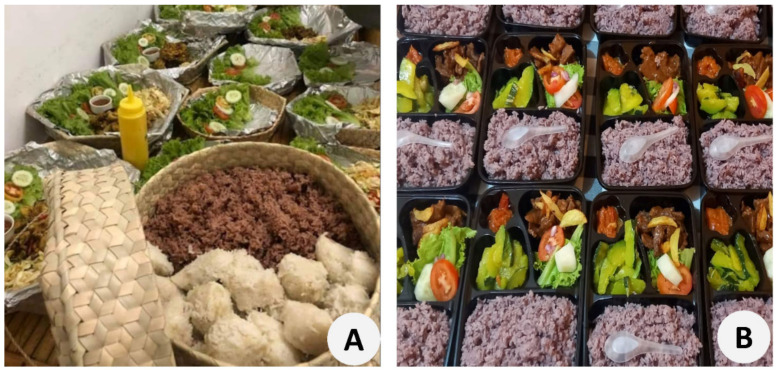
Traditional meals prepared with bean species cultivated in association with cashew agroforestry systems in East Timor: (**A**) Ceremonial meal with *Vigna unguiculata* (black) at the forefront; (**B**) traditional meal with rice, vegetables, and *Vigna unguiculata* from Baucau district (Photos L. Guterres and M.N. Ximenes).

**Table 1 foods-11-03503-t001:** Studied bean species: place of collection, scientific name, number of samples, main seed colour, and code.

Location	Coordinates	Altitude (m)	Taxon ^1^	No. of Samples	Main Seed Colour ^1^	Code ^2^
Batugadé	−8.966868° 124.966229°	45	*Cajanus cajan* (a)	5	Cream	CCa.C
				5	Black	CCa.Bl
				5	Brownish	CCa.Br
				6	Cream and brown	CCa.CBr
Beseuc	−9.432905° 125.108694°	7	*Cajanus cajan* (b)	5	Cream	CCb.C
				5	Black	CCb.Bl
				3	Brownish	CCb.Br
				5	Cream and brown	CCb.CBr
Kefamenanu	−9.441529° 124.483582°	426	*Phaseolus lunatus* (a)	10	Purple and white	PLa.PW
				1	Cream and purple	PLa.CP
				4	Brownish	PLa.Br
Fatucahi	−9.037613° 125.971621°	51	*Phaseolus lunatus* (b)	10	Purple and white	PLb.PW
				1	Cream and purple	PLb.CP
				4	Brownish	PLb.Br
Ai-oan	−9.410409° 125.147416°	7	*Phaseolus vulgaris*	5	Brown	PV.Br
				5	Brown	PV.Br
				5	Brown	PV.Br
Baucau	−8.487492° 126.370399°	513	*Vigna angularis*	5	Bordeaux	VA.Bo
				5	Bordeaux	VA.Bo
				5	Bordeaux	VA.Bo
Fatucahi	−9.037613° 125.971621°	51	*Vigna radiata*	5	Green	VR.G
				5	Green	VR.G
				5	Green	VR.G
Fatucahi	−9.021160° 125.964844°	51	*Vigna unguiculata* (a)	5	Bordeaux	VUa.Bo
				5	Bordeaux	VUa.Bo
				5	Bordeaux	VUa.Bo
Fatucahi	−9.037613° 125.971621°	51	*Vigna unguiculata*(b)	5	Black	VUb.Bl
				5	Black	VUb.Bl
				5	Black	VUb.Bl

^1^ Samples of the same species but from different locations are referred as (a) and (b). ^2^ Code: species and colour morphotype.

**Table 2 foods-11-03503-t002:** Legume species present in cashew agroforestry systems of East Timor. For each species: provenance (I—introduced; N—native), habit (T—tree; S—shrub or subshrub; C—climber), common names (Mc, Macassae—indigenous language of Baucau; Km, Kemak—indigenous language of Bobonaro; Tt, Tetum; Ml, Malayu), surveyed location; and main uses.

Species	Status	Habit	Common Name (Language)	Location ^1^	Food	Fodder/Forage	Medicine	Domestic Uses
*Acacia ancistrocarpa* Maiden & Blakely	I	T	Ai-kasi (Tt)	Bobonaro (Batugadé); Manatuto (Cribas, Manatuto)		●		●
*Acacia auriculiformis* A.Cunn. ex Benth.	N?	T	Ai-kasi, ai-bubur mutin (Tt)	Cova Lima (Ai-oan); Manatuto (Cribas)		●		●
*Acacia dealbata* Link	I	T	Ai-kafe malae, funan kinur (Tt)	Bobonaro (Sanirin); Manatuto (Natarbora)		●	●	
*Acacia iteaphylla* F.Muell. ex Benth.	I	T	Ai-tasi tarak, funan kinur (Tt)	Bobonaro (Sanirin); Manatuto (Cribas, Manatuto)		●		
*Acacia mangium* Willd.	I	T	Ai-bubur (Tt)	Manatuto (Cribas)			●	●
*Acacia stenophylla* A.Cunn. ex Benth.	I	T	Ai-kasi, tarak funan kinur (Tt)	Bobonaro (Batugadé, Sanirin); Manatuto (Manatuto)		●	●	
*Afzelia quanzensis* Welw.	I	T	Musan matan mean (Tt)	Manatuto (Natarbora)				●
*Albizia julibrissin* Durazz.	I	T	Samtuku malae (Tt)	Bobonaro (Batugadé)		●		●
*Alysicarpus vaginalis* (L.) DC.	I	S	Duút (Tt)	Manatuto (Cribas, Natarbora)		●	●	
*Arachis hypogaea* L.	I	S	Fore-rai (Tt)	Bobonaro (Batugadé)	●	●	●	
*Arachis pintoi* Krapov. & W.C.Greg.	I	S	Fore-fuik	Manatuto (Cribas, Natarbora)			●	
*Bauhinia cheilantha* (Bong.) Steud.	I	T		Manatuto (Cribas)			●	
*Caesalpinia pulcherrima* (L.) Sw.	I	T	A-funan tarak, Ai-manu-ikun	Manatuto (Natarbora)	●	●	●	
*Cajanus cajan* (L.) Huth (=*Cajanus indicus* Spreng.)	I	S	Tunis, Tunis makerek, metan, mutin (Tt)	Bobonaro (Batugadé) [Also in: Aileu (Aileu); Ainaro (Maubisse); Baucau (Gariuri, Triloca, Venilale); Cova Lima (Beseuc, Suai, Suai Loro); Dili (Ataúro-Makili); Ermera (Gleno, Railaco); Liquiçá (Liquiçá); Manatuto (Cribas)]	●	●	●	
*Calopogonium mucunoides* Desv.	I	C	Lehe mutin fuik (Tt)	Manatuto (Cribas)	●	●	●	
*Canavalia gladiata* (Jacq.) DC.	I	C	Koto-moruk (Tt)	Cova Lima (Beseuc)	●	●		
*Cassia javanica* Vell.	N	T	Laiki (Mc); Ai-kasi moruk (Tt)	Manatuto (Cribas)		●	●	
*Centrosema pubescens* Benth.	I	C	Koto fuik (Tt), Fore-mungo fuik (Tt)	Bobonaro (Batugadé)		●		
*Chamaecrista nictitans* (L.) Moench	N	S	Du’ut maria moe dor (Tt)	Manatuto (Manatuto); Viqueque (Viqueque)		●		
*Crotalaria pallida* Aiton	I	S	Kalaur fuik (Tt)	Bobonaro (Maliana-Maumali)		●	●	
*Delonix regia* (Bojer ex Hook.) Raf.	I	T	Ai-kasi funan mean (Tt)	Bobonaro (Sanirin); Manatuto (Cribas, Natarbora)	●	●		●
*Erythrina abyssinica* Lam.	I	T	Ai-dik funan mean (Tt)	Manatuto (Cribas)			●	
*Erythrina americana* Mill. (=*Erythrina coralloides* Moc. & Sessé ex DC.)	I	T	Ai-dik malae funan mean (Tt)	Bobonaro (Batugadé)			●	
*Erythrina fusca* Lour.	I	T	Ai-dik malae funan mean (Tt)	Bobonaro (Batugadé)			●	
*Erythrina lysistemon* Hutch.	I	T		Manatuto (Natarbora)			●	
*Erythrina speciosa* Andrews	I	T	Ai-dik funan mean (Tt)	Bobonaro (Batugadé); Manatuto (Natarbora)			●	
*Gliricidia sepium* (Jacq.) Kunth	I	S	Amare (Tt); Gamal (Ml)	Baucau (Triloca); Bobonaro (Batugadé, Maliana-Maumali, Sanirin); Lautém (Lospalos); Manatuto (Cribas, Natarbora)	●	●	●	●
*Glycine max* (L.) Merr.	I	S	Fore-keli	Bobonaro (Sanirin)	●	●	●	
*Indigofera suffruticosa* Mill.	I	S	Ai-Taun (Tt), daru (Mc)	Bobonaro (Batugadé, Maliana-Maumali, Sanirin); Manatuto (Cribas, Natarbora)				●
*Lathyrus oleraceus* Lam. (=*Pisum sativum* L.)	I	S	Ervilha (Mc)	Manufahi (Fatucahi)	●	●		
*Leucaena leucocephala* (Lam.) de Wit	I	T	Ai-kafe Timor (Tt)	Baucau (Baucau); Bobonaro (Batugadé, Sanirin); Manatuto (Cribas)	●	●	●	●
*Mimosa pudica* L.	I	S	Duút Maria Moe dor (Tt)	Bobonaro (Maliana-Maumali); Manatuto (Cribas)			●	
*Moringa oleifera* Lam.	I	T	Marungi (Tt)	Bobonaro (Sanirin); Manatuto (Cribas, Natarbora); Manufahi (Fatucahi)	●		●	
*Mucuna pruriens* (L.) DC.	I	C	Lehe metan (Tt)	Manatuto (Cribas, Natarbora)	●	●	●	
*Pachyrhizus erosus* (L.) Urb.	I	T	Sinkumas (Tt)	Bobonaro (Batugadé); Manatuto (Cribas)	●		●	
*Phaseolus lunatus* L.	I	C	Koto moruk (Tt) Koto mean	West Timor (Indonesia), Kupang District (Kefamenanu) [Also in: Baucau (Baguia, Venilale); Covalima (Suai); Manufahi (Fatucahi, Same)]	●	●	●	
*Phaseolus vulgaris* L	I	C	Koto mean (Tt)	Manufahi (Fatucahi) [Also in: Baucau (Baguia, Fatumaca, Quelicai, Venilale); Aileu (Liquidoe, Seloi); Ainaro (Ainaro, Hatu Builico, Maubisse); Cova Lima (Ai-oan)]	●	●		
*Samanea saman* (Jacq.) Merr. (=*Albizia saman* (Jacq.) F.Muell.)	I	T	Ai-matan dukur (Tt)	Bobonaro (Maliana-Maumali, Tunu-bibi); Manatuto (Cribas)		●	●	●
*Senna alata* (L.) Roxb.	I	S	Senna	Manatuto (Natarbora)			●	
*Senna occidentalis* (L.) Link	I	S		Bobonaro (Maliana-Maumali)	●	●	●	
*Senna siamea* (Lam.) H.S.Irwin & Barneby (=*Cassia siamea* Lam.)	I	T	Ai-kaixote ou ai-kasi (Tt)	Bobonaro (Sanirin); Manatuto (Cribas)	●	●	●	●
*Senna tora* (L.) Roxb. (=*Cassia tora* L.)	I	S	Fore-rai fuik (Tt), Bibu hure (Km)	Bobonaro (Sanirin); Manatuto (Cribas)	●	●	●	
*Sesbania grandiflora* (L.) Poir.	I	S	Ai-turi (Tt)	Baucau (Baucau); Bobonaro (Sanirin); Manufahi (Fatucahi)	●	●	●	●
*Styphnolobium japonicum* (L.) Schott	I	T	Ai-kaixote, Ai-Kasi funan mutin (Tt)	Manatuto (Cribas, Natarbora)			●	
*Tamarindus indica* L.	I	T	Sukaer	Bobonaro (Batugadé, Sanirin)	●	●	●	●
*Vachellia farnesiana* (L.) Wight & Arn	I	T	Xira (Mc)	Bobonaro (Batugadé, Sanirin); Manatuto (Cribas); Manufahi (Fatucahi)		●		
*Vigna angularis* (Willd.) Ohwi & H.Ohashi	I	C	Fore-masin (Tt)	Baucau (Baucau); Cova Lima (Ai-oan, Beseuc, Suai) [Also in: Aileu]	●	●		
*Vigna mungo* (L.) Hepper	I	C	Fore-mungo metan	Cova Lima (Suai)			●	
*Vigna radiata* (L.) R.Wilczek	N	C	Fore-mungo (Tt)	Cova Lima (Ai-oan) [Also in: Liquiçá (Liquiçá); Cova Lima (Suai); Manufahi (Betano, Fatucahi, Same)]	●	●	●	
*Vigna unguiculata* (L.) Walp.	I	C	Foretali mean; Foretali metan (Tt)	Baucau (Baucau); Bobonaro (Batugadé); Cova Lima (Ai-oan); Manufahi (Fatucahi) [Also in: Aileu (Aileu); Ainaro (Ainaro, Maubisse); Baucau (Baguia, Baucau, Bercoli, Bucoli, Buruma, Gariuri, Laga, Quelicai, Seiçal, Triloca, Uailili, Venilale); Dili (Ataúro); Ermera (Gleno, Railaco); Lautém (Lospalos); Liquiçá (Liquiçá); Manatuto (Laclubar, Manatuto); Manufahi (Fatucahi); Oecussi-Ambeno (Naimeco, Padiae)]	●	●	●	

^1^ Only locations where the legume species are part of the agroforestry system are referred. For the six bean species studied, other areas where they are commonly grown are within square brackets.

## Data Availability

All related data and methods are presented in this paper. Additional inquiries should be addressed to the corresponding author.

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
