# Peer review of "Diversity of Legumes in the Cashew Agroforestry System in East Timor (Southeast Asia)"

_foods, 2022, doi:10.3390/foods11213503_

Round 1

Reviewer 1 Report

The paper titled " Diversity of legumes in the cashew agroforestry system in East Timor (Southeast Asia)" deals within the scope of the Journal. In this study, the authors systematically described the situation of legume intercropping in cashew agroforestry system in East Timor, analyzed the diversity and mineral composition of some cereal legumes, and also discussed the sustainable utilization of legumes. Field surveys conducted in the context of this study revealed the diversity of legumes associated with cashew agroforestry systems and provided a reference for the people of East Timor on micronutrient deficiencies. Overall, the manuscript is more detailed, and the methods are technically feasible. In addition, the English of the manuscript should be proofread and polished. Considering the novelty, implications, experimental design, and data analysis of the manuscript, in my opinion, the manuscript needs the following revisions:

Detailed comments are listed as follows:

- "Abstract" section needs to be a detailed, concise, and logical description of the findings of the article, please keep your language simple.

- Line 23-24: Please describe the main purpose of your research in detail and completely.

- Line 25: Please specify the range of these fifty species, are all beans?

- "Introduction" section of the text formatting, literature citation exists many issues, this is not allowed, please revised.

- Line 48-49: Confirm the authenticity of the data and the comparison range of GDP.

- Line 53-54: Please modify the expression of "the first three quarters of the 20th century".

- Line 88-89: The tense of this sentence should be revised.

- The methodology section must be improved. The description must allow reproducibility and comprehension by the readers. Please enhance the details and description.

- Line 122-125: Confirm the accuracy of the data in this section and the relationship between the total area and the area of each crop.

- Line 125: Is the area of cashew cultivation data accurate? Taken together, cashew should be widely grown locally.

- Line 132: Please check the format is correct, "2.2" should only have a pause after the number.

- Line 134-153: Please explain the differences between the two agricultural systems and the sampling work therein, and whether they are legumes that are symbiotic with cashew orchards.

- Line 185-199: References of the experimental methods should be added.

- "Results" section of the reference basis for serious insufficient, control to add.

- Line 241-243: Please explain the purpose and significance of the five plants reported by the local residents.

- Line 246-249: Please add the references on which to base.

- Line 291-293: Please add the references on which to base.

- Line 302: Why does this plant adapt well to acidic soil? Please add your scientific basis.

- Line 307-308: Add your scientific evidence.

- Line 332-335: The tense of this sentence should be revised.

- "Discussion" section added less objective analysis, please revised.

- Line 486-488: Discuss toxicity and add references.

- Line 500-502: Please add the references on which to base.

- Line 519-522: Please add the references on which to base.

- Conclusion needs to be improved with a specific and concise description of the results obtained.

- Line 568-570: The tense of this sentence should be revised and preferably add specific data support.

Author Response

REVIEWER 1:

The paper titled "Diversity of legumes in the cashew agroforestry system in East Timor (Southeast Asia)" deals within the scope of the Journal. In this study, the authors systematically described the situation of legume intercropping in cashew agroforestry system in East Timor, analyzed the diversity and mineral composition of some cereal legumes, and also discussed the sustainable utilization of legumes. Field surveys conducted in the context of this study revealed the diversity of legumes associated with cashew agroforestry systems and provided a reference for the people of East Timor on micronutrient deficiencies. Overall, the manuscript is more detailed, and the methods are technically feasible. In addition, the English of the manuscript should be proofread and polished. Considering the novelty, implications, experimental design, and data analysis of the manuscript, in my opinion, the manuscript needs the following revisions:

Response to reviewer:

Dear Reviewer,

We are grateful for the time you dedicated to review this manuscript, and for the very positive comments you have given. Below are the point-by-point responses to your comments:

Q1. "Abstract" section needs to be a detailed, concise, and logical description of the findings of the article, please keep your language simple.

Response to reviewer: We recognize that this is an important comment, also mentioned by Reviewer 2. We believe that in the new version of the manuscript, the addition of new details fully addresses the concerns of both Reviewers by providing more information about the objectives, methodologies, and results. We have also tried to make the language simpler and more concise (see Lines 20-34).

Q2. Line 23-24: Please describe the main purpose of your research in detail and completely

Response to reviewer: We very much appreciate this helpful comment and some alterations have been incorporated in the revised version of the manuscript. Lines 105-109: “The present work has three main goals: (i) to characterise the diversity of legume species intercropping in the cashew agroforestry systems of East Timor and to evaluate their multiple uses; (ii) to analyse the diversity and mineral composition (macro and microminerals) of some grain legumes (beans); and (iii) to discuss the sustainable use of legumes in the context of food security in East Timor.”

Q3. "Introduction" section of the text formatting, literature citation exists many issues, this is not allowed, please revised.

Response to reviewer: We recognize that these issues were not sufficiently addressed in the previous version of our study. So, we have performing some modifications in the Introduction section, and we add or reformulated some paragraphs (47-50; 53-55; 87-96) to support our findings. More recent references (e.g. 3; 23; 24; 25) were added.

Q4. Line 48-49: Confirm the authenticity of the data and the comparison range of GDP.

Response to reviewer: We agree with the Reviewer and the data has been updated with recent information accessed at World Bank concerning the GDP regarding to agriculture share. As such, the text has been modified as follows: “Since its independence in 2002, East Timorese main activities have been agriculture, fishing, and forestry, and the country's agriculture share of GDP decreased from 25.6% in 2010 to 15.4% in 2020 [3]”. A new reference was added to support this sentence [3]:      “World Bank. Timor-Leste: Share of economic sectors in the gross domestic product (GDP) from 2010 to 2020. Available online at https://www.statista.com/statistics/728875/share-of-economic-sectors-in-the-gdp-in-timor-leste/ [Accessed 18 October 2022]”. (Lines: 47-50)

Q5. Line 53-54: Please modify the expression of "the first three quarters of the 20th century".

Response to reviewer: The sentence was corrected accordingly. We are very grateful to the Reviewer for his meticulous reading of the paper.

The text now reads: “The current agricultural landscapes of the country primarily result from the food production mechanisms implemented by the Portuguese in the 20th century, who promoted rice in coastal areas and coffee at higher altitudes until the 1970s [5].” (Lines: 53-55)

Q6. Line 88-89: The tense of this sentence should be revised.

Response to reviewer: We have considered this comment by the Reviewer, and the new version of the manuscript was checked for typological, grammatical, and spelling errors.

Q7. The methodology section must be improved. The description must allow reproducibility and comprehension by the readers. Please enhance the details and description.

Response to reviewer: We recognize that some of these issues were not sufficiently addressed in the previous version of our study. We now have made changes to the methodology section, for instance:

2.1. Studied area (Lines: 128-132); 2.2. Diversity of legumes in cashew agroforestry systems - 2.2.1. Sampling (Lines 137-150; and 156-160); 2.3. Mineral analyses (Lines: 183-186); 2.4. Statistical analysis (Lines: 205-211).

Q8. Line 122-125: Confirm the accuracy of the data in this section and the relationship between the total area and the area of each crop.

Response to reviewer: We recognize that some of these issues were not sufficiently addressed in the previous version of our study. We would like to state that the accuracy of the data has been confirmed, with total area and area of each crop being updated with recent data obtained from area harvested at FAO Statistical database for Timor-Leste. Thus, the section has been modified according to reviewer comment at L, as follows:

According to the East Timor Agriculture Census 2019 [33] and to FAOSTAT 2022 data [34], agricultural holdings in the country cover a total area of 216,189 ha, the top three productions being coffee 36,626 ha, maize 29,455 ha and rice 20,719 ha. Cashew occupies about 800 ha and bean crops about 10,505 ha, with Phaseolus vulgaris being one of the most cultivated (9152 ha) and Phaseolus lunatus one of the least cultivated (264 ha)” (Lines 128-132).

Q9. Line 125: Is the area of cashew cultivation data accurate? Taken together, cashew should be widely grown locally.

Response to reviewer: We would like to clarify that the present study provides first scientific survey covering cashew orchards of East Timor. Therefore, we believe that the area of cashew cultivation remains to be properly accessed in East Timor. In fact, an estimated 800 ha reported for cashew cultivation area is from the late 1990s census. Nevertheless, during an integrated and extensive fieldwork on cashew agroecosystem developed in East Timor, an expansion of cashew orchards was depicted in areas where other crops were previously cultivated, yet proper assessment of cashew cultivation area in East Timor is still lacking, considering the increasing trend of cashew establishment as an industrial crop.

Q10. Line 132: Please check the format is correct, "2.2" should only have a pause after the number.

Response to reviewer: We have considered this comment by the Reviewer, and the new version of the manuscript was checked.

Q11. Line 134-153: Please explain the differences between the two agricultural systems and the sampling work therein, and whether they are legumes that are symbiotic with cashew orchards.

Response to reviewer: We agree with the Reviewer that the section "2.2. Legume diversity in cashew agroforestry systems - 2.2.1. Sampling” section was confusing and subdivided into different points that became confusing. Therefore, in order to clarify the sampling done by the first author in East Timor, the section 2.2.1 was rewritten and we believe it became much clearer (see Lines 137 - 160)

Q12. Line 185-199: References of the experimental methods should be added.

Response to reviewer: We agree with the Reviewer and new references were added in the new version of the manuscript, namely:

Catarino, L., Romeiras, M.M., Bancessi, Q., Duarte, D., Faria, D., Monteiro, F., & Moldão, M. Edible leafy vegetables from West Africa (Guinea-Bissau): Consumption, trade and food potential. Foods 2019, 8(10), 493. doi:10.3390/foods8100493” and

“Catarino, S., Brilhante, M., Essoh, A.P., Charrua, A.B., Rangel, J., Roxo, G., et al. Exploring physicochemical and cytogenomic diversity of African cowpea and common bean. Sci. Rep. 2021, 11(1), 1-14. doi:10.1038/s41598-021-91929-2” (Lines 190-193)

Q13. "Results" section of the reference basis for serious insufficient, control to add.

Response to reviewer: We recognize that some issues were not sufficiently addressed in the Results section. So, we have performing some modifications, and more recent references we add to support our findings, namely: Lines 244 – 254; 261-271.

Q14. Line 241-243: Please explain the purpose and significance of the five plants reported by the residents.

Response to reviewer: According to the Reviewer suggestion more details were added on the significance of the five reported species. Lines 244- 254: “Gliricidia sepium produces edible flowers and the leaves are an important food source for cattle. The species is frequently harvested for firewood and is also used in traditional medicine [35,43]. Leucaena leucocephala is mainly used for cattle food, but the seeds could also be used for cooking [44]. Senna siamea fruits and leaves can be eaten as a vegetable; it is also used as a shade tree, forage, and medicinal plant [43]. Sesbania grandiflora is used as firewood, house building, and food in rural communities. It is an important species to feed the ruminants, as medicine, organic fertilizer and ornamental plant [35; 44]. Tamarindus indica produces tasty fruits very employed in human consumption and its timber is proper for house building [35]. This species is suitable to feed animals and for medicinal purposes against diarrhea, constipation, fever and other diseases [44].”

Q15. Line 246-249: Please add the references on which to base.

Response to reviewer: We agree with the Reviewer and three references were added in the new version of the manuscript. We include the reference to three databases widely used to check the main uses of the species, namely: “Plants of the World Online. http://powo. science.kew.org [Accessed March 17, 2022]”; “Plant Resources of Tropical Africa. https://www.prota4u.org/database/ [Accessed March 17, 2022].” and “Useful Tropical Plants Database. https://tropical.theferns.info/ [Accessed March 17, 2022].”

Q16. Line 291-293: Please add the references on which to base.

Response to reviewer: We agree with the Reviewer and a reference has been incorporated in the revised version of the manuscript. The text now reads: “The great distribution and diversity of kidney bean demonstrate that it is extremely well adapted to the climatic conditions of East Timor since this is one of the most climatically demanding bean species [45].” Reference 45 - Line 304.

Q17. Line 302: Why does this plant adapt well to acidic soil? Please add your scientific basis.

Response to review:  We would like to clarify that a parallel study aiming the identification of soils types has been done in the sampled cashew orchards. Although this study is not yet published for East Timor, and we prefer in this stage to remove this sentence.

Q18. Line 307-308: Add your scientific evidence.

Response to reviewer: We agree with the Reviewer and a reference has been incorporated in the revised version of the manuscript

Q19. Line 332-335: The tense of this sentence should be revised.

Response to reviewer: We have considered this comment by the Reviewer, and the new version of the manuscript was checked for typological, grammatical, and spelling errors.

Q20. - "Discussion" section added less objective analysis, please revised. Line 486-488: Discuss toxicity and add references.

Response to reviewer: We agree with the Reviewer and two references were added in the revised version of the manuscript (Lines 520-526):

  1. Dangi, P., Chaudhary, N., Gajwani, D. Antinutritional Factors in Legumes. In: Handbook of Cereals, Pulses, Roots, and Tu-bers. CRC Press, 2021. p. 305-318.
  2. Ekanayake, S., Skog, K., Asp, N. G. Canavanine content in sword beans (Canavalia gladiata): Analysis and effect of processing. Food Chem Toxicol, 2007, 45(5), 797-803.

Q21. Line 519-522: Please add the references on which to base.

Response to reviewer: We agree with the Reviewer and two references were added in the revised version of the manuscript. Lines 557-560: “Iron is required to transfer oxygen to the body's tissues and organs, and anaemia is the most common nutritional deficiency, affecting more than 2 billion people worldwide [62]. Zinc plays an essential role in human metabolism, improving the immune system and preventing disease [63].”

  1. Abbaspour, N., Hurrell, R., Kelishadi, R. Review on iron and its importance for human health. . J Res Med Sci 2014; 19:164-74.
  2. Maares, M., Haase, H. Zinc and immunity: An essential interrelation. Arch. Biochem. Biophys, 2016, 611, 58-65. doi.org/10.1016/j.abb.2016.03.022

Q22. Conclusion needs to be improved with a specific and concise description of the results obtained.

Response to reviewer:  We agree with the Reviewer and in the new version, the conclusion section has been rewritten in order to reflect the main achievements of this study, summarizing its results and it is highlighted that the studied legume species are excellent sources of minerals, representing an invaluable potential to meet the nutritional needs of the Timorese populations (see Lines: 590-610).

Q23. Line 568-570: The tense of this sentence should be revised and preferably add specific data support.

Response to reviewer: We have considered this comment by the Reviewer, and the new version of the manuscript was checked for typological, grammatical, and spelling errors.

We look forward to hearing from you at your earliest convenience.

Maria Romeiras, on behalf of all the authors

Reviewer 2 Report

Authors present the assessment of legumes diversity in the cashew agroforestry system in East Timor. The manuscript is appropriately written, and it contains relevant contributions to a sustainable strategy for reducing micronutrient deficiencies. However, the manuscript could be improved after the following revisions:

Abstract:

It would be interesting to briefly present methods of the work and to briefly present how results sustain the main conclusions observed. It would also be interesting to prevent the contributions of this study.

Introduction:

Authors says, from line 83, that “… Legumes are among the primary sources of human food, being essential components of a healthier diet. Nutritionally, they are 2-3 times richer in protein than cereals, and they also provide significant amounts of minerals...” Although the importance of legumes on human diet is described, it would be interesting to discuss it with more details, showing what minerals are expected and how they could contribute to human health.

Methods:

It would be interesting to add references of methods that are without it (for example, mineral analyses, from line 184 to 202).

For ANOVA (analysis of variance), it would be interesting to present whether experiments were conducted at random sequence and the number of replications for each treatment.

Results:

It would be interesting to add if ANOVA assumptions were guaranteed (homogeneity of variance, independent residuals and normal distribution of residuals).

Conclusions:

It would be interesting to rewrite conclusions to a more specific text related to the main goals of this study. It would be interesting to present the main results and discussions that are related to these goals, showing how these results contributes to a sustainable strategy for reducing micronutrient deficiencies.

Author Response

REVIEWER 2.

Authors present the assessment of legumes diversity in the cashew agroforestry system in East Timor. The manuscript is appropriately written, and it contains relevant contributions to a sustainable strategy for reducing micronutrient deficiencies. However, the manuscript could be improved after the following revisions:

Dear Reviewer,

Thank you very much for your valuable comments and suggestions on our manuscript. Below are the point-by-point responses to your comments:

Q1. Abstract: It would be interesting to briefly present methods of the work and to briefly present how results sustain the main conclusions observed. It would also be interesting to prevent the contributions of this study.

Response to reviewer:  We recognize that this is an important comment, also mentioned by Reviewer 2. We believe that in the new version of the manuscript, the addition of new details fully addresses the concerns of both Reviewers by providing more information about the objectives, methodologies, and results. We have also tried to make the language simpler and more concise. (Lines 20-34).

Q2. Introduction: Authors says, from line 83, that “… Legumes are among the primary sources of human food, being essential components of a healthier diet. Nutritionally, they are 2-3 times richer in protein than cereals, and they also provide significant amounts of minerals... ” Although the importance of legumes on human diet is described, it would be interesting to discuss it with more details, showing what minerals are expected and how they could contribute to human health.

Response to reviewer: We recognize that some of these issues were not sufficiently addressed in the previous version of our study. We now have made changes to the text, for instance we gave more details about the minerals present in legumes and their expected contribution to human health. New references were added to support the paragraphs (see Lines 87-96).

Q3. Methods: It would be interesting to add references of methods that are without it (for example, mineral analyses, from line 184 to 202). For ANOVA (analysis of variance), it would be interesting to present whether experiments were conducted at random sequence and the number of replications for each treatment.

Response to reviewer:  We have carefully considered this comment by the Reviewer, and so we have changed the Methods section providing new references. We would like to refer that not all data followed a normal distribution, therefore we replaced the parametric tests by non-parametric tests. Also, the number of replications were added to the methods (see Lines: 183-186).

Q4. Results: It would be interesting to add if ANOVA assumptions were guaranteed (homogeneity of variance, independent residuals, and normal distribution of residuals).

Response to reviewer: We agree with the Reviewer and the proposed alterations were all considered. In fact, not all our mineral contents followed a normal distribution, therefore we homogenized our statistical analysis by applying non-parametric tests to all our data, Mann-Whitney, Kruskall-Wallis and the post Hoc Dunn test. With the new results the homogeneous groups, we have performed some modifications and the Figures 5, 6, 7, 8 were updated. Also, the Supplementary files - Table S1 was updated according to new statistical tests (see Lines: 205-211).

Q5. Conclusions: It would be interesting to rewrite conclusions to a more specific text related to the main goals of this study. It would be interesting to present the main results and discussions that are related to these goals, showing how these results contributes to a sustainable strategy for reducing micronutrient deficiencies.

Response to reviewer:  We agree with the Reviewer and in the new version, the conclusion section has been rewritten in order to reflect the main achievements of this study, summarizing its results and it is highlighted that the studied legume species are excellent sources of minerals, representing an invaluable potential to meet the nutritional needs of the Timorese populations (see Lines: 590-610).

We look forward to hearing from you at your earliest convenience.

Maria Romeiras, on behalf of all the authors